# Mutual information: Measuring nonlinear dependence in longitudinal epidemiological data

**Alexander L. Young** [1]☺*, **Willem van den Boom**[2‡], **Rebecca A. Schroeder**[3‡],
**Vijay Krishnamoorthy**[3‡], **Karthik Raghunathan**[3‡], **Hau-Tieng Wu**[4‡], **David B. Dunson**[4,5☺]

**1** Department of Statistics, Harvard University, Cambridge, Massachusetts, United States of America, **2** Yong Loo Lin School of Medicine, National University of Singapore, Singapore, Singapore, **3** Department of Anesthesiology, Duke University, Durham, North Carolina, United States of America, **4** Department of Mathematics, Duke University, Durham, North Carolina, United States of America, **5** Department of Statistical Science, Duke University, Durham, North Carolina, United States of America

☺ These authors contributed equally to this work.
‡ WB, RAS, VK, KR, and HTW also contributed equally to this work
* alexander_young@fas.harvard.edu

**Data Availability Statement:** The data used for this study has been anonymized and made available through the Duke University Library Digital

## Abstract

Given a large clinical database of longitudinal patient information including many covariates, it is computationally prohibitive to consider all types of interdependence between patient variables of interest. This challenge motivates the use of mutual information (MI), a statistical summary of data interdependence with appealing properties that make it a suitable alternative or addition to correlation for identifying relationships in data. MI: (i) captures all types of dependence, both linear and nonlinear, (ii) is zero only when random variables are independent, (iii) serves as a measure of relationship strength (similar to but more general than $R^2$), and (iv) is interpreted the same way for numerical and categorical data. Unfortunately, MI typically receives little to no attention in introductory statistics courses and is more difficult than correlation to estimate from data. In this article, we motivate the use of MI in the analyses of epidemiologic data, while providing a general introduction to estimation and interpretation. We illustrate its utility through a retrospective study relating intraoperative heart rate (HR) and mean arterial pressure (MAP). We: (i) show postoperative mortality is associated with decreased MI between HR and MAP and (ii) improve existing postoperative mortality risk assessment by including MI and additional hemodynamic statistics.

## Introduction

In modern epidemiology, it is common to consider datasets having many variables collected over time on each study participant. This includes health monitoring data and environmental exposure data for both healthy individuals and patients undergoing a procedure. In such contexts, it is important to have statistical tools available for quantifying dependence between different variables. In this article, we motivate the use of Mutual Information (MI) as a general metric for measuring the (non)linear dependence between variables. To demonstrate its utility, we show how MI can be used to improve extant epidemiological models with a brief example

Repository for Research Data at DOI: 10.7924/
r45q52g2t (https://doi.org/10.7924/r45q52g2t).

**Funding:** ALY was supported by the National
Science Foundation (Award #1045153, Award 331
#1546130). WvdB was supported by the National
Institute of Environmental Health 332 Sciences
(NIEHS) of the National Institutes of Health (NIH)
(R01-ES017240). The funders had no role in study
design, data collection and analysis, decision to
publish, or preparation of the manuscript.

**Competing interests:** The authors have declared
that no competing interests exist.

on the association between postoperative mortality and intraoperative blood pressure and
heart rate, both variables which are modifiable by clinicians. In this setting, we demonstrate a
clinically relevant connection between postoperative mortality and the nonlinear dependence
between HR and MAP as measured by MI.

MI, however, is not a commonly used statistic in many statistical analyses. The $R^2$ value,
which corresponds to the square of the correlation coefficient, is the standard choice for a sin-
gle summary statistic of the strength of the statistical association between two variables. The $R^2$
statistic is appealing in being equal to zero (up to statistical uncertainty due to limited sample
sizes) for independent random variables and increasing as the degree of linear dependence
between variables increases. It is also easy to calculate, and the interpretation is familiar to
epidemiologists.

However, $R^2$ inherits the disadvantages of the correlation coefficient in focusing exclusively
on the level of linear dependence between variables. Nonlinear dependence is extremely com-
mon in epidemiology, and in such cases the correlation coefficient may underrepresent the
degree of dependence between different variables. A potential solution is to view one variable
as a response and the other variable as a predictor, fit a flexible nonlinear regression model,
and then report the regression $R^2$ value as a measure of dependence. This modified approach
is sensitive to which variable is treated as the outcome and can also be sensitive to the nonlin-
ear regression model used in the analysis. This approach is often ad hoc and risks overfitting
which can greatly inflate the estimated dependence.

Given this issue, there has been considerable recent interest on the development of novel
statistical methods for the detection of dependence (linear or nonlinear) with a great emphasis
on statistical power [1]. Examples include the Maximal Information Coefficient (MIC) [2, 3],
Hilbert-Schmidt Independence Criterion [4], distance correlation (dCor) [5], and the Ran-
domized Dependence Coefficient [6]. A 2015 empirical study suggested that the Total Infor-
mation Coefficient, an extension of MIC, is state of the art in the settings tested with regards to
power against independence. However, those numerical and theoretical results pertaining to
MIC have been contested in other studies [7, 8].

Herein, we use the estimated dependence in downstream models. In our study, we had
paired observation of between 240 and 500 HR and MAP measurements for a cohort of
patients. For each patient, we wished to estimate the strength of the dependence between their
HR and MAP, which was subsequently used to assess postoperative mortality risk. In this set-
ting, power to detect dependence was less important than a statistic which could be estimated
with minimal error from a moderate sample size, thereby enabling reliable comparisons
between patients. As such, we elected to use mutual information (MI), which was estimated
using the Kraskov-Stögbauer-Grassberger (KSG) nearest neighbor method [9]. An empirical
study justifying this choice may be found in S1 File. This study shows that KSG outperforms
MIC and dCor with regards to i) mean squared error under independence and ii) mean
squared error and relative error under dependence for the sample sizes considered in our
study on HR and MAP.

MI is a useful and elegant statistic capturing all types of dependence between random quan-
tities [9, 10] and can supplement analyses based on $R^2$ or other summary statistics when inves-
tigating dependency in epidemiologic data. In a series of studies on the early prediction of
sepsis from clinical data, Nesaragi et al [11–13] developed a tensor factorization method for
assessing risk of infection within a six-hour window. Their method used raw observations and
statistical summaries including pairwise dependency measures to identify elevated risk of sep-
sis. In the three separate articles, using MI as the measure of dependency [13] in their tensor
factorization method outperformed the same algorithm when using ratios and powers [11] or
correlations [12]. Despite this example, MI is not routinely used in the study of

epidemiological data. As such, a short exposition to build intuition on MI is also provided in the subsequent Methods Section to provide the epidemiologic community with greater insight and intuition on this statistic. Throughout this paper, we use $I(X, Y)$ to denote the MI of random variables $X$ and $Y$.

In the remainder of the article, we discuss a clinically relevant application of MI in a retrospective cohort study of 35,709 surgical procedures conducted in the Duke University Health System from 1999 to 2003 which is the primary result and focus of this article. In this study, we investigate the association of hemodynamic summary statistics with postoperative mortality, a topic of considerable recent interest [14–20] given the ability to modify perioperative hemodynamic variables and potentially improve clinical outcomes in this population. Intraoperative HR and MAP measurements were aligned at 30 second intervals. The means and standard deviations of these quantities and their correlation coefficient and MI were calculated for each patient. To assess the association of $I(HR, MAP)$ with postoperative mortality, two statistical models were used: Cox Proportional Hazards (CPH) and Logistic Regression (LR). In each case, additional demographic and surgical covariates were included, which are discussed later. The CPH analysis demonstrates a clear association between $I(HR, MAP)$ and postoperative mortality even after adjusting for other variables including the correlation coefficient of HR and MAP. The correlation of HR and MAP shows no association with mortality. This is a concrete example of a case where MI provides information that $R^2$ does not. Additionally, the inclusion of hemodynamics including $I(HR, MAP)$ is shown to improve existing postoperative risk assessment scoring algorithms based on LR.

## Methods

### Mutual information: Properties and interpretation

In this section, we provide an illustrative description of MI. For readers interested in the mathematical formulation of MI, please see S1 File which contains mathematical definitions in terms of the probability mass functions (pmfs) and probability density functions (pdfs) in the case of discrete and continuous variables, respectively. A self-contained review of these topics can also be found in [21]. All MI estimates were obtained using the k-nearest neighbor algorithm proposed by [9]. The details of this estimator are also provided in S1, and it has been implemented in R functions available in a code repository.

Generically, given a choice of independent and dependent variables, the $R^2$ value indicates the proportion of the variability in the dependent variable that is accounted for by the independent variable. In this setting, $R^2 = 0$ ($R^2 = 1$) indicates that none (all) of the variability in the dependent variable can be explained by the independent variable. However, as noted in the introduction, the value of $R^2$ is sensitive to the model chosen to relate the two quantities. In the simple case of linear regression, which we assume to be the case hereafter, $R^2$ is the square of the correlation coefficient between two variables and is unaffected by the choice of independent and dependent variables. Unfortunately, if the linear relationship between the data is weak, $R^2$ will be small even when the data exhibit a strong, highly predictive nonlinear relationship.

As a preliminary, motivating example, let us consider the risk factor data obtained from www.gapminder.org including the average rates of alcohol consumption, body mass index, mean systolic blood pressure (SBP), cholesterol, and proportion of the population that smokes. Each datum corresponds to averages of the male population in different countries taken in the year 2005. In Fig 1, we show scatterplots of these data including lines of best fit along with the estimated $R^2$ values between each pair of variables. In particular, note the very weak linear relationships between SBP and both smoking rate ($R^2 = 0.00197$) and cholesterol ($R^2 = 0.0392$).

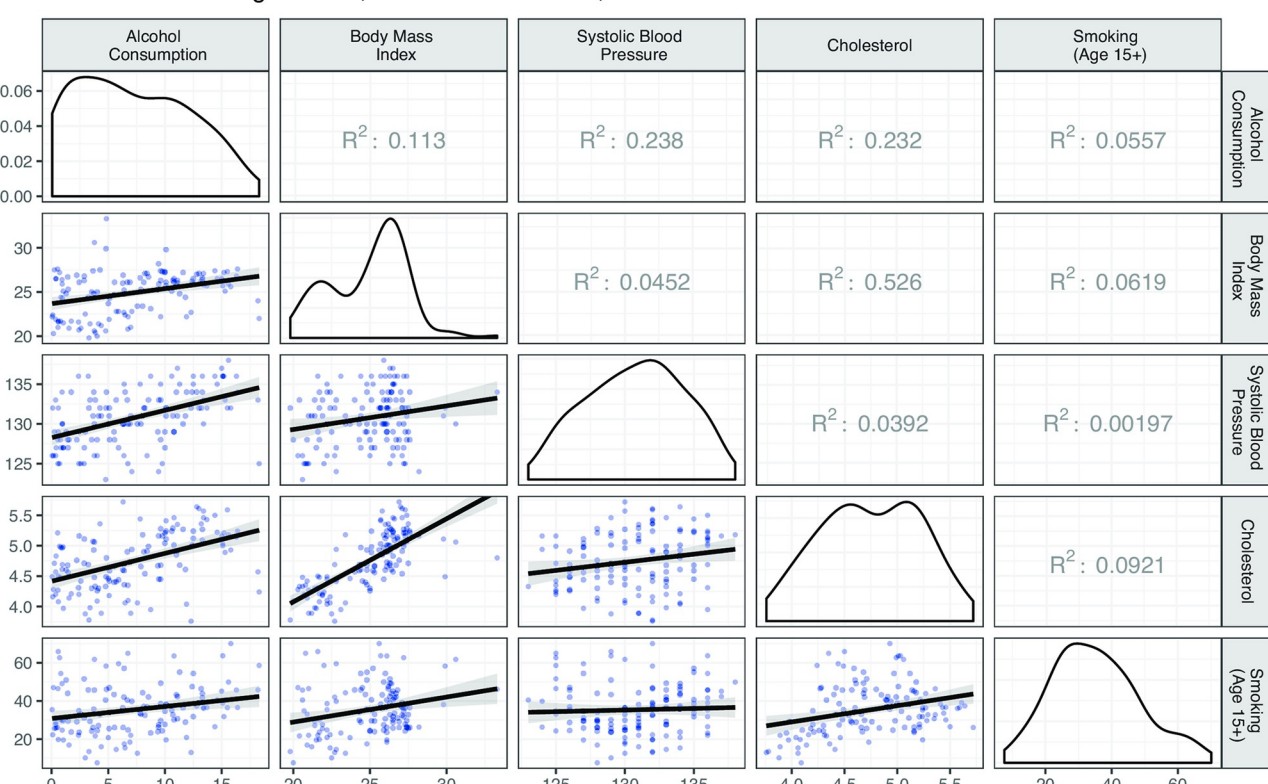

**Fig 1. Estimated densities for the five risk factors in the gapminder data are shown along the diagonal.** Below the diagonal, pairwise scatterplots of the data including linear regression (black lines) with 95% confidence intervals (shaded grey regions) are provided. Above the diagonal, the $R^2$ values, indicating the percent of variation accounted for by the linear relationships between the data, are provided. Source: multiple sources for the individual variables aggregated by www.gapminder.org.

However, upon visual examination, there does appear to be some nonlinear dependence between some of the variables. In particular, note the parabolic relationship between SBP and smoking.

Alternatively, one can take the following view; each observation of a random variable provides information about the patient from which it was sampled. The amount of information is measured in units of bits where, informally, a bit represents the answer to a yes/no question. If we observe two random variables, it is natural to expect that we attain more information about the patient; however, some of that information is shared or redundant between the observations. Importantly, when the variables are strongly (weakly) associated, linearly or nonlinearly, one can expect the amount of shared information to be large (small). In this perspective, we would like to estimate the MI between X and Y, which is a statistic measuring the average amount of shared information over repeated observations of X and Y. Formally, if $I(X, Y) = 1$ bit, then, on average, an observation of $X$ ($Y$) provides the answer to one yes/no question about $Y$ ($X$). Importantly, $I(X, Y) = I(Y, X)$ so that $X$ provides as much information about $Y$ as $Y$ does about $X$. Importantly, this perspective holds if $X$ and $Y$ are continuous, discrete, or mixed random variables. Mutual information does share one important property with $R^2$. Both measures are unchanged by linear transformations to the random variables including standardization.

In Fig 2, we show fits, obtained using smoothing splines, between the variables in the gapminder dataset. In all cases except SBP and cholesterol, the fitting algorithm detects nonlinear

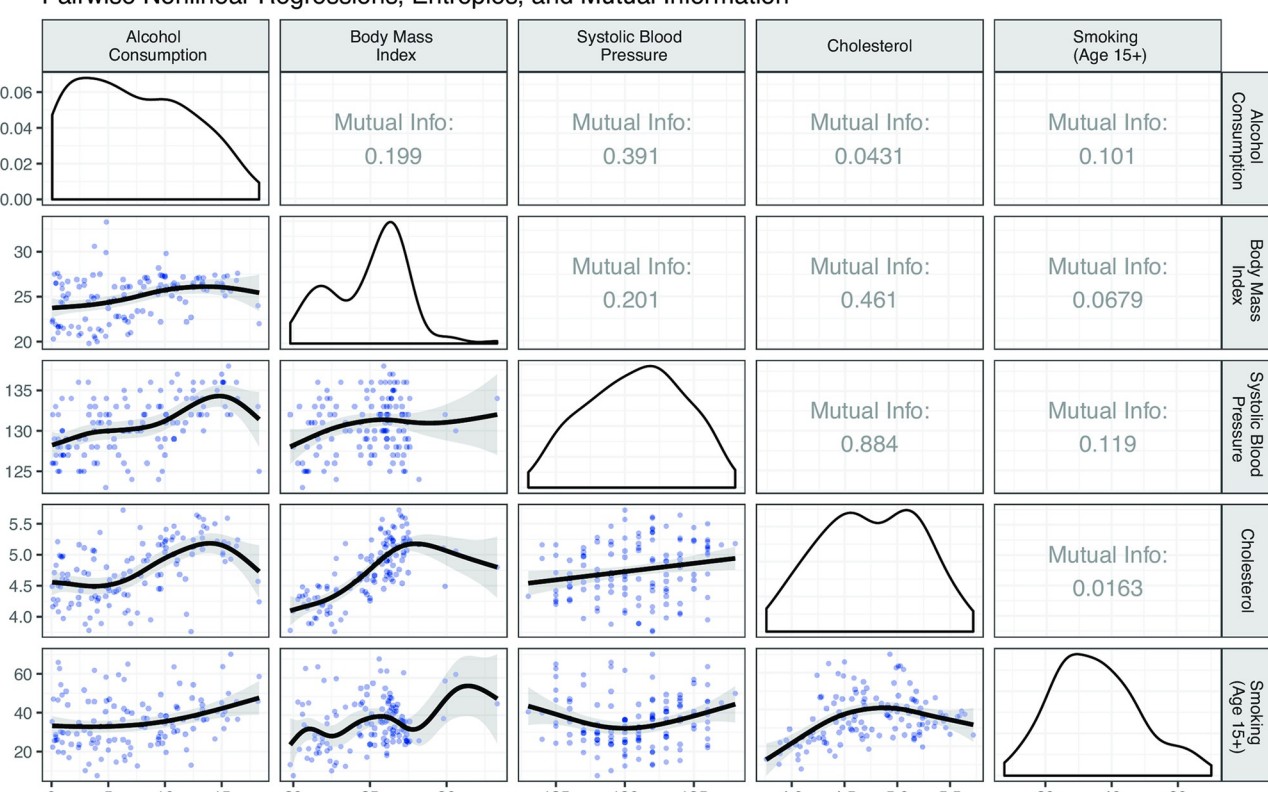

**Fig 2. Pairwise plots of the risk factor variables are again provided but with nonlinear smoothing spline fits.** The fits are shown as black lines with 95% confidence intervals as shaded grey regions. Above the diagonal, the estimated pairwise mutual information between the variables in bits are provided. These values indicated the average amount of shared information between the observations. All mutual information values were obtained using k-nearest neighbors type estimators (see S1) with k = 20.

relationships suggesting that the correlation coefficient is an inadequate measure of the association in the dataset. Alternatively, we show estimates of the MI between each pair of variables in the upper portion of Fig 2. Using MI, we conclude that smoking and cholesterol have the weakest association, 0.0163 bits of shared information, whereas SBP and cholesterol have the strongest association, 0.887 bits of shared information. In this case, MI was estimated using the KSG nearest neighbor method with $k = 20$. Some of the results in Fig 1 are sensitive to this choice of $k$. For example, taking $k = 10$ gives an estimate of $I(SBP, cholesterol) \approx 0.322$ which is moderate when compared to other pairwise MI estimates. Using $k = 10$, cholesterol and BMI again have the strongest estimated associate with an MI of 0.703 bits. Complete results for the estimates using $k = 10$ are provided in S1 File. Optimal method for choosing $k$ with the KSG estimator is an open question.

It is natural to wonder if there are guidelines for determining whether MI is small or large; that is, how to determine the strength of dependence of various kinds. MI is a measurement of information, so it is nonnegative. Importantly, if I(X, Y) = 0, then there is no shared information between X and Y, which can occur if and only if X and Y are independent. Thus, an MI near zero is considered small. Unfortunately, unlike correlation there is not an a priori upper bound to MI. In general, it is not straightforward to develop guidelines for cutoffs of MI indicating weak or strong relationships. Some rescaled versions of MI have been proposed to address this issue [21]. For technical reasons, all such approaches are only suitable in the case

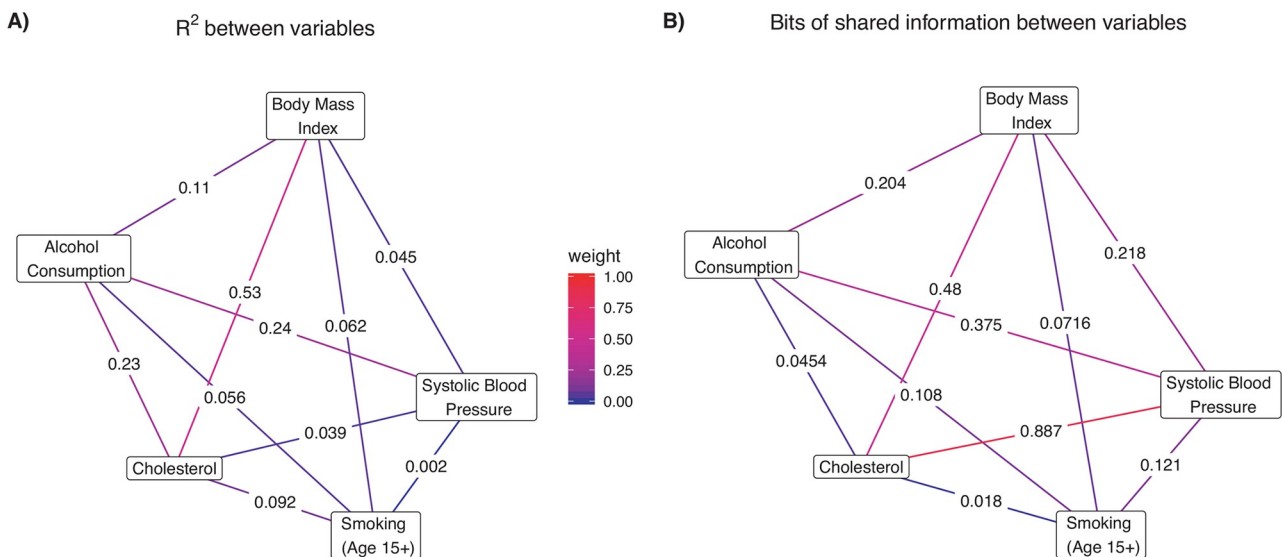

**Fig 3. The dependence between the risk factor variables captured via correlation using $R^2$ values are shown in the left image.** The right image shows the dependence using mutual information measured in bits. The mutual information estimates were obtained using k-nearest neighbors type estimators (see the appendix) with k = 20.

of discrete data [21] and do not apply in the case of continuous data, and hence are inappropriate for the data considered in this article. This issue, which is an inherent weakness of MI in comparison to $R^2$, is discussed further in S1 File.

As a guideline, we suggest that an MI between two variables be considered small or large relative to the MIs of other variable pairs. From this perspective, using MI to compare pairwise dependency is equivalent to asking which pairs of variables contain the largest amount of shared information. This a very normal approach to data analysis. For example, differences in population means are considered large if they differ by multiple standard deviations. However, this does not give an absolute scale for comparing differences. Correlation is essentially unique in having an absolute scale that can be easily interpreted.

Returning to the gapminder data, Fig 3 shows network plots with edge weights indicating the strength of the association, using the correlation coefficient (Left) and MI (Right). Notably, SBP and cholesterol shared the most information with a MI of 0.887 bits indicating the strongest association in the data. This result provides a fundamentally different indication than the $R^2$, which suggested that SBP and cholesterol had the second weakest relationship.

## Intraoperative hemodynamics and mortality

Turning to our primary example, we consider the association of intraoperative hemodynamics with post-operative mortality in the retrospective cohort study mentioned in the introduction.

## Patient characteristics and database

The Duke Medicine Institutional Review Board for Clinical Investigations approved this study (approval no.: Pro0067683). After full review, the requirement for informed consent of this retrospective study was waived as the data were analyzed anonymously. The raw data for this study contains partial records for 431,480 surgical cases at Duke University Hospital and has been used in other studies examining surgical mortality [22, 23].

Of interest here are the 79,985 cases with detailed intraoperative hemodynamic measurements. In the database, 35,709 (44%) cases contained records with sufficient data for this

study. In particular, records with incomplete demographic data or missing ASA Status or Emergency codes (14,517 cases, 18%) or ASA Status code equal to 5 (349 cases, <1%) were excluded. The latter exclusion was chosen to limit the focus of our analyses to cases where surgery was not required by the severity of the patient's medical status. ASA Status six is reserved for deceased patients and is omitted here as well. Furthermore, I(HR,MAP) estimates are known to be biased for small sample sizes [9, 24, 25]. As such, cases with fewer than 240 HR/MAP measurements were also excluded (29,410 cases, 37%). The final study cohort of 35,709 procedures was then stratified into patients who underwent cardiac procedures and patients who underwent noncardiac procedures using ICD-9 codes. The cardiac cohort contained 10,457 (29.3%) procedures, and the noncardiac cohort contained 25,252 (70.7%) procedures. The reporting of this observational study adheres to the applicable STROBE guidelines [26].

## Code and data availability

The data used for this study has been anonymized and made available through the Duke University Library Digital Repository for Research Data doi: 10.7924/r45q52g2t. Code for estimating mutual information may be found in a github repository.

## Variables

In the original raw data, the time-stamped HR and MAP data were quantized to integer values and unpaired. These time-stamped data were recorded at 30-second intervals and saved separately. In some cases, small errors resulted in shifts in the time-stamps. To align the observations, the time-stamps for all HR and MAP recordings for a patient were rounded to the nearest 30 seconds and then paired if entries with a matching stamp existed.

The number of HR/MAP measurements was included in the CPH model as a covariate, as the length of the procedure may have an impact. Additionally, as the original data are quantized to integer values, independent, uniform jitters from -0.5 to 0.5 were added to each data point prior to estimating the MI. The jitters, which were required for technical requirements of the MI estimator, may be viewed as proxies for the missing decimals lost when the data were quantized. For each case, 25 separate, independent realizations of the jitters were generated. The MI estimates were averaged to obtain the final MI estimate used in the CPH and LR studies. The variability was less than 5% of the mean value across all patients and realizations of the jitters. In all cases, $I(HR, MAP)$ was estimated using the KSG method with $k = 5$. This value of $k$ was chosen for to balance faster estimation with minimal sampling variability as data were jittered.

In addition to I(HR, MAP), which we view as the exposure of interest, the means, standard deviations, and absolute value of the correlation, $|Corr(HR, MAP)|$, of the paired HR and MAP data were calculated. Using the signed correlation does not alter the results substantively. Patient age and the number of HR/MAP estimates for each patient were also used as summary statistics in the subsequent models. Furthermore, the patients' ASA status and indication of an emergency procedure were included as covariates. For comparison with existing methodologies, our LR analysis also included the risk level of the procedure, measured as low, medium, or high, for the noncardiac cases. Unlike the CPH study, noncardiac cases with unclassified risk were excluded as were all cardiac cases, which are inherently high risk. Again, these exclusions were made for comparison with existing methodologies. Summaries of the numerical (Table 1) and categorical predictors (Table 2) are provided subdivided by procedure type.

In both analyses our outcome of interest was patient mortality. In the CPH study, we focused on time-to-death following surgery whereas the LR study considered 30-day mortality. To ascertain mortality, records were linked to the Duke Decision Support Repository.

**Table 1. Summaries of the statistics pertaining to the age and hemodynamic variables.**

| Covariate, median (IQR) | Variable | Cardiac Procedures (n = 10,457) | Noncardiac Procedures (n = 25,252) |
|---|---|---|---|
| Age, years | $X_1$ | 64 (54 –72) | 61 (50- 70) |
| Number of records | $X_2$ | 498 (416–598) | 516 (374–683) |
| Mean HR, bpm | $X_3$ | 76.2 (70.2–83.7) | 73.3 (65.4–83) |
| Mean MAP, mmHg | $X_4$ | 74.7 (70.4–79.3) | 83.5 (76.9–92) |
| Std. Dev. HR, bpm | $X_5$ | 14.5 (11.4–18.1) | 8.4 (6.3–11.2) |
| Std. Dev. MAP, mmHg | $X_6$ | 15.7 (12.9–18.9) | 14.4 (11.6–17.8) |
| $|Corr(HR, MAP)|$, | $X_7$ | 0.84 (0.78–0.90) | 0.89 (0.76–0.94) |
| I(HR,MAP) | $X_8$ | 2.4 (2.2—2.6) | 2.3 (2.1—2.6) |

## Multivariable cox proportional hazards model of postoperative mortality

The outcome of interest is time to death following the surgical procedure, which is a right-censored outcome. All cases where the patient survived more than one year were censored. To assess the association of I(HR, MAP) and other hemodynamic summary statistics with postoperative mortality, a Multivariable Cox Proportional Hazards (CPH) model was used.

The CPH regression analysis followed a generalized additive structure. The continuous covariates contributing to the relative change in the hazard function were modeled as unknown nonlinear functions. Indicators of ASA status and Emergent operations were included as linear predictors, which resulted in a hazard function of the following form:

$$\lambda(t) = \lambda_0(t)\exp\left(\sum_{j=1}^{8} f_j(X_{ij}) + \sum_{j=9,10,12,13} b_j X_{ij}\right)$$

where $\lambda_0(t)$ is the baseline hazard function. Here $X_i = (X_{i,1}, \ldots, X_{i,13})$ are the covariates for patient $i$. Note that ASA Status 3 and Nonemergent Procedures were used as references, and thus excluded from the covariate list. A complete summary of these covariates is provided in Tables 1 and 2. The CPH was fit to these partially-censored time to event data in the cardiac

**Table 2. Summaries of categorical covariates pertaining to ASA status, emergent or nonemergent operation, and risk of surgical procedure.** ASA Status is a numerical score from one to six of overall patient health from one (healthy patients) to five (critically ill patients). Cases with an ASA Code of five were excluded as there were too few such cases for reliable analysis.

| Indicator | Variable | Cardiac Procedures (n = 10,457) | Noncardiac Procedures (n = 25,252) |
|---|---|---|---|
| ASA Code | | Counts | Counts |
| 1 | $X_9$ | 3 (0%) | 270 (1%) |
| 2 | $X_{10}$ | 238 (2.2%) | 5,847 (23%) |
| 3 | $X_{11}$ | 4,026 (39%) | 15,909 (63%) |
| 4 | $X_{12}$ | 6,190 (59%) | 3,226 (13%) |
| Emergency Status | | Counts | Counts |
| Emergency | $X_{13}$ | 596 (6%) | 1,968 (8%) |
| Nonemergency | $X_{14}$ | 9,861 (94%) | 23,284 (92%) |
| Surgical Risk | | | Counts |
| Low | $X_{15}$ | N/A | 2,849 (11%) |
| Medium | $X_{16}$ | N/A | 3,779 (15%) |
| High | $X_{17}$ | N/A | 9,808 (39%) |
| Unclassified | | N/A | 8,816 (35%) |

and noncardiac cohorts separately using the 'mgcv' package version in R version 3.2.3. This includes nonlinear fits for the functions of $f_1, \ldots, f_8$. The analysis was performed separately for the cardiac and noncardiac procedures.

### Logistic regression for mortality prediction in noncardiac procedures

[27] constructed the Surgical Mortality Probability Model based on a LR modeling using only ASA Status, Emergent/Nonemergent Operations, and a classification of procedure risk as low, medium, or high. None of the indicators in that model are modifiable intraoperatively. It is natural to extend their model by including hemodynamic variables which could motivate intervention guidelines for anesthesiologists in the future.

The outcome of interest is 30-day mortality in noncardiac procedures. Note, the outcome of interest in this study is survival, a binary result, differs from the outcome of interest in the CPH study, which focused on time to death, a continuous outcome. We extend the work in [27] by adding hemodynamic variables to the indicators from Table 2 in a LR model of the following form:

$$P(Death_i \mid X_i) = logit^{-1}\left(\sum_{j=3-8,10-12,16,17} b_j X_{i,j}\right).$$

We elected to exclude age and the number of HR/MAP measurements as they are nonmodifiable intraoperatively, and we wish to focus on hemodynamics alone. Indicators of ASA Status 1, nonemergency, and low risk procedures were used as references and excluded from the model as well. The noncardiac data were split into equal-sized training and test subsets. Both models were calibrated on the training data. The model incorporating hemodynamic data was then compared to the more parsimonious model of [27] using C-statistics computed from the test data. This process was repeated 400 times to estimate the standard error in the C-statistics.

## Results

### Clinical and demographic characteristics

The complete cohort studied contained 35,709 cases. Of these, 15,414 (43%) patients were female and 20,295 (57%) were male. The cohort of cardiac procedures consisted of 10,457 (29%) cases whereas 25,252 (71%) cases were for noncardiac procedures. Summaries of the covariates included in the CPH and LR for these two cohorts may be found in Tables 1 and 2. There were 3,793 (11%) deaths recorded. Within the cardiac cohort, 197 (1.9%) patients died within 30 days of their surgical procedures and 672 patients (6.4%) died within one year of surgery. Within the cohort of noncardiac procedures, 574 (2.3%) patients died within 30 days of surgery and 3,121 (12.4%) died within a year of surgery. For 16,436 cases with a surgical risk classification of low, medium, or high used in the LR study, there were 310 (1.2%) deaths within 30 days of the surgical procedure.

### Distribution of $I(HR, MAP)$

In Fig 4, we show the distribution of I(HR, MAP) over the entire study population subdivided by cardiac and noncardiac procedures. In both cohorts, $I(HR, MAP)$ ranges from 0 to 5 bits. Based on visual inspection, we suggest rough cutoffs of $I(HR, MAP) \geq 4$ bits for large MI and $I(HR, MAP) \leq 2.5$ for small MI.

The implication of these cutoffs is most apparent if one considers sample hemodynamics with extremal values of MI. In Fig 5, we show the HR and MAP time series for two different

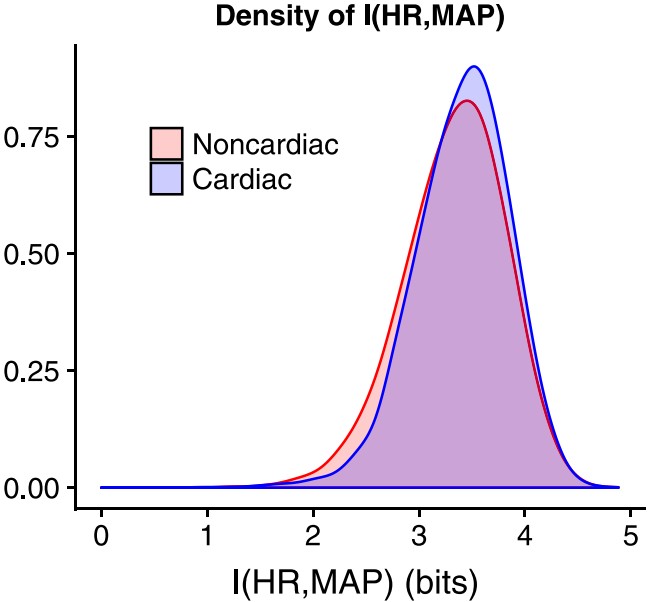

**Fig 4. The distribution of patient I(HR,MAP) is shown for the cohorts of patients undergoing cardiac and noncardiac procedures.** We will consider a patient's I(HR,MAP) to be large (small) if it is greater than (less than) approximately 4 bits (2.5 bits).

patients who were chosen from the population because their correlation of intraoperative HR and MAP was very small.

In Fig 5A, large fluctuations in the patient's MAP occurred with no apparent association to the HR, which remained nearly constant throughout the procedure. For this patient, *I(HR, MAP)* = 2.1 bits indicates a weak association relative to the other patients in the cohort. In Fig 5B, the coupling of HR and MAP is striking upon visual inspection suggesting a strong relationship between HR and MAP, which is reflected by the MI estimate of *I(HR, MAP)* = 4 bits. To understand the clinical implications of the MI of HR and MAP, we next discuss the methods used to examine these relationships using retrospective cohort study design.

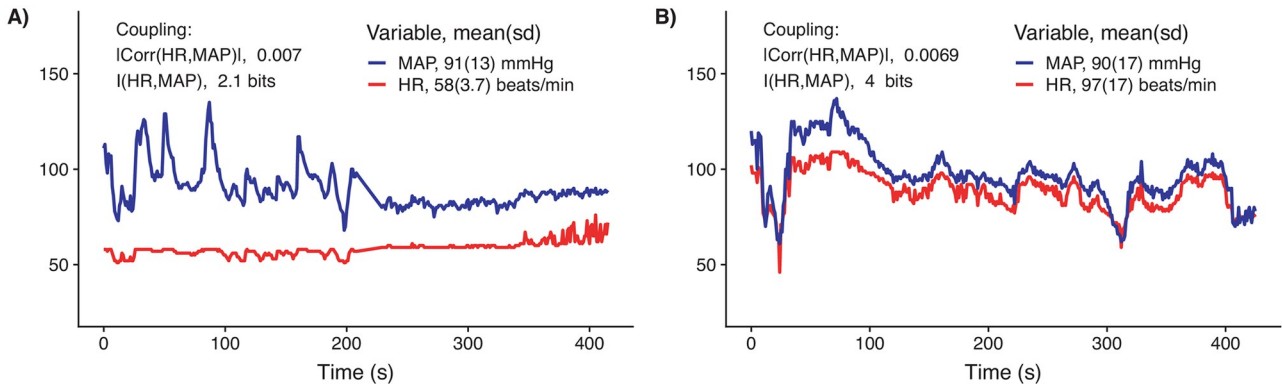

**Fig 5. Two examples of patient HR and MAP data are shown above.** (A) The patient's HR remains roughly constant throughout the procedure. The variability in MAP does not have a clear association with HR which is consistent with a small *I(HR, MAP)* = 2.1 bits. (B) In this case, HR and MAP appear much more tightly coupled and exhibit similar dynamics throughout the procedure which is reflected in a large *I(HR, MAP)* = 4 bits. The correlation coefficient of HR and MAP, shown in the figures, were small and nearly equal which may wrongly suggest that HR and MAP were similarly dependent in these two cases.

## Results of CPH

The primary interest of this study is the determination of covariates with the most clinically-relevant association with mortality. Means and 95% confidence intervals are reported for each linear predictor in Table 3.

Fitted functions with means and 95% confidence intervals are presented for covariates involving HR and MAP in Fig 6.

For the number of measurements, see Fig 7.

The largest increases in the Hazard Ratio across both surgery types were associated with increased Mean HR in both cardiac and noncardiac procedures (Fig 6G and 6H) and decreased Mean MAP in cardiac procedures (Fig 6E and 6F). However, within the hemodynamic covariates, I(HR,MAP) was the only other covariate beyond mean HR with a consistent relationship with mortality across both cohorts (Fig 6A and 6B). While correlation would traditionally be used to assess dependence between HR and MAP, the magnitude of the association between HR/MAP correlation and mortality is smaller than I(HR,MAP) (Fig 6C and 6D) and the relationship is very different between the two surgery types. The same observation holds for the number of HR/MAP measurements. These results demonstrate clear advantages to using $I(HR, MAP)$ in postoperative risk assessment beyond univariate summary statistics or correlation. Further, as HR and MAP decouple and approach independence, mortality risk greatly increases, an observation which may be of interest to anesthesiologists for potential targets of intraoperative intervention.

## Results of logistic regression

Of the 25,252 noncardiac procedures, the risk of 16,436 procedures were classified as low, medium, or high, as outlined in [27]. To assess performance of the two models, the 16,436 procedures were randomly split into equal sized training and test cohorts of 8,218 patients. Each model was fit on the training portion of the data, and AUCs were then obtained from ROC curves based on predictions of the test portion of the data. This process was repeated 400 times to attain the mean and standard error for the AUC estimates. The results are summarized in Table 4.

First, we note the AUC reported here is lower than reported in the original manuscript [27]. This is perhaps not surprising given the limitations of research using electronic medical records (EMR). However, including the additional hemodynamic covariates results in clear improvements (C = 0.749) over the Glance model (C = 0.708). While the difference in performance is not drastic, our results suggest that the inclusion of the given hemodynamic

**Table 3. The mean and 95% confidence intervals of the hazard ratio associated with each of the linear indicators in the CPH.** The outcome of interest in this study was the time to death were right censored for death occurring more than 365 from the surgical procedure.

| Covariate | Cardiac Procedures (n = 10,457) | | Noncardiac Procedures (n = 25,252) | |
|---|---|---|---|---|
| ASA Code | Mean | CI | Mean | CI |
| 1 | N/A | | 0.14 | 0.04, 0.43 |
| 2 | 0.3 | 0.07, 1.2 | 0.47 | 0.41, 0.53 |
| 3 | Ref. | | Ref. | |
| 4 | 2.2 | 1.8, 2.7 | 1.7 | 1.6, 1.9 |
| Emergency Status | | | | |
| Emergency | 1.8 | 1.4, 2.3 | 0.97 | 0.86, 1.1 |
| Nonemergency | Ref. | | Ref. | |

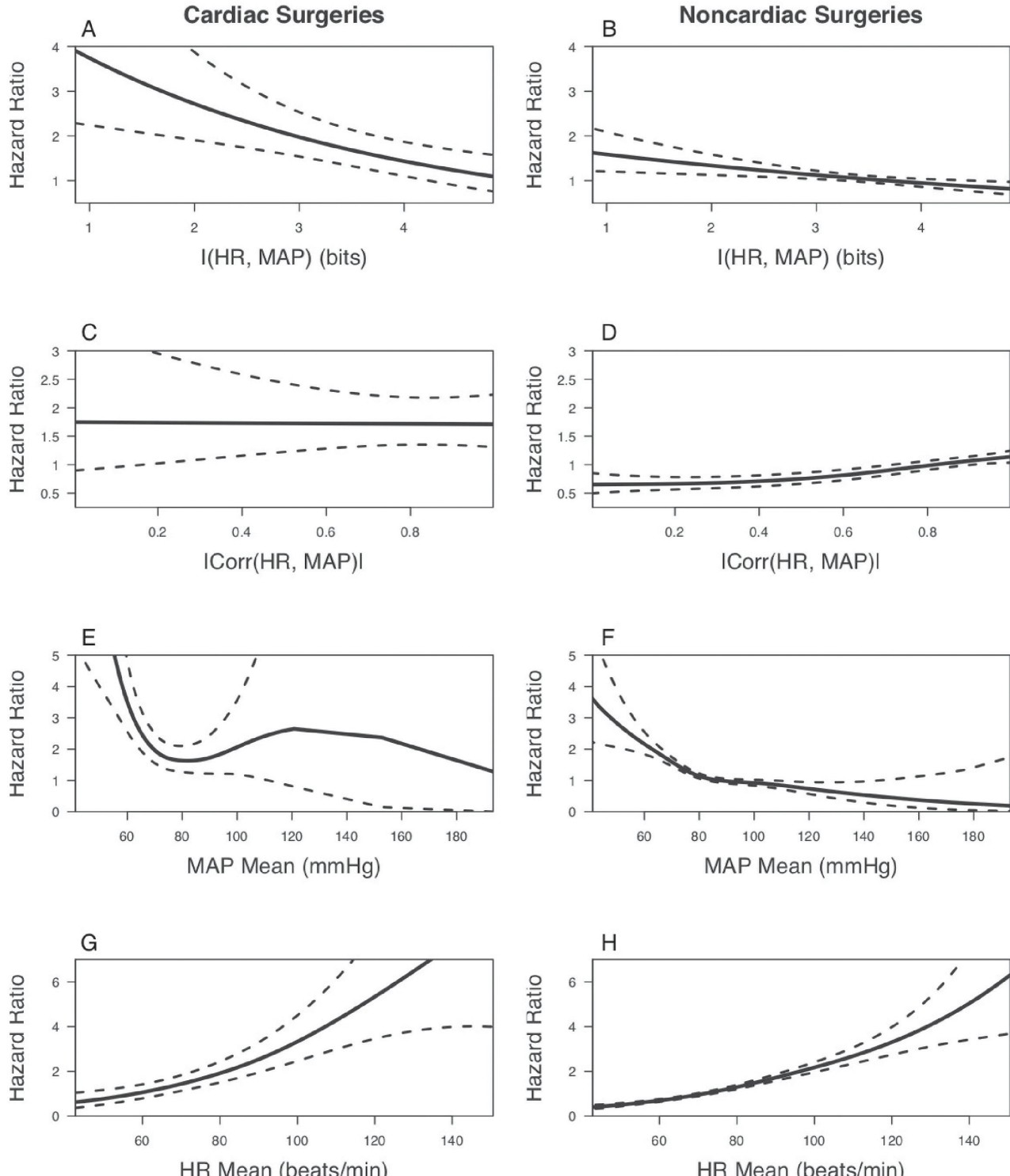

**Fig 6. The hazard ratio from the CPH is shown as a function covariates involving HR and MAP.** Mean estimates are shown as solid lines with 95% confidence intervals as dashed lines. The most clinically relevant associations with mortality are elevated mean HR, decreased mean MAP, and decrease I(HR, MAP).

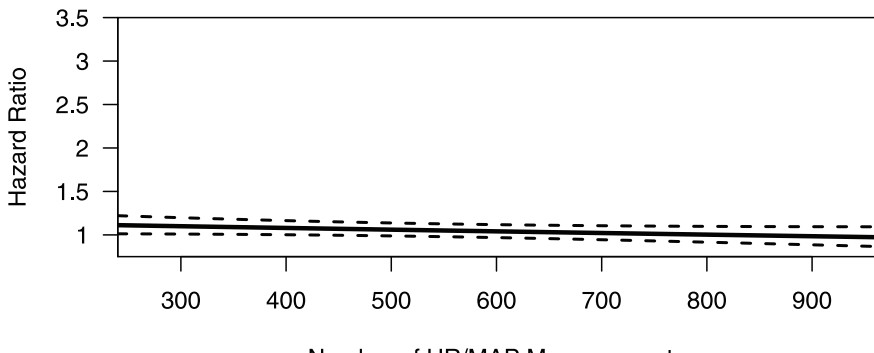

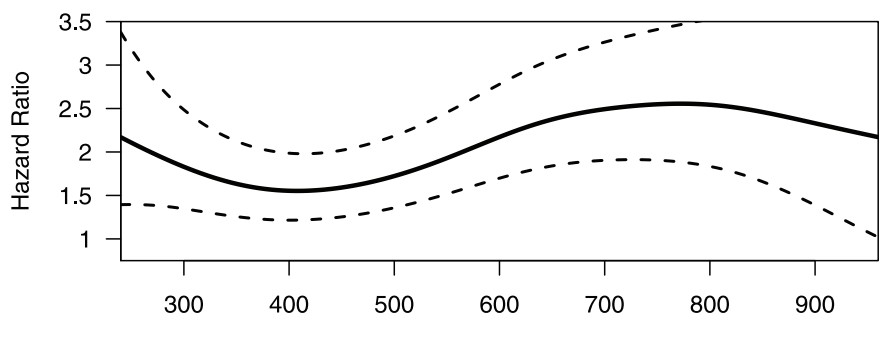

**Fig 7. The hazard ratio from the CPH is shown as a function of the number of HR/MAP measurements.** The mean estimate is shown as a solid line with 95% confidence intervals as dashed lines.

summary statistics does improve predictive performance. Importantly, as HR and MAP are routinely recorded during surgical procedures, they could be easily included in postoperative risk assessment algorithms. Doing so can provide a small but inexpensive and simple improvement to existing algorithms [27] in practice.

## Discussion

MI captures all types of dependence in data offering clear advantages to $R^2$. In the preliminary study of risk factors aggregated by www.gapminder.org, the results using MI suggested a number of relationships were stronger than predicted by $R^2$, which also failed to capture the nonlinear relationships in the data. In particular, the relationship between SBP and cholesterol was the strongest coupling of variables in the data when viewed through MI. This is a stark contrast to the results of $R^2$ which indicated SBP and cholesterol were very weakly dependent.

**Table 4. AUCS from ROC curves from the logistic regression models.** The outcome of interest in this model was death within 30 days of the surgical procedure.

| Model | [27] | [27] with Hemodynamics |
|---|---|---|
| AUCs, mean (SE) | 0.708 (0.0008) | 0.749 (0.0007) |

Beyond the ability of MI to capture nonlinear dependence in data, we also highlighted its added utility in being symmetric (similar to correlation) and interpreted identically for categorical and numerical data. These advantages stem from viewing epidemiologic data in the context of (redundant) information in data rather than variability. As such, MI serves as a supplement to correlation when assessing dependency within a dataset.

Finally, the retrospective study of surgical outcomes demonstrated a clear application of MI as a predictor with strong clinical implications. The integration of heart rate and blood pressure has shown strong associations with mortality in previous studies [14–16, 20] our investigation of MI within hemodynamic data shows that it is feasible to estimate I(HR,MAP) using routinely collected intraoperative data, and this quantity is strongly associated with postoperative mortality. Furthermore, the addition of hemodynamics including coupling in existing risk assessment models [27] results in improved performance. ASA Status, Emergency Codes, and age were strongly associated with postoperative mortality in the CPH for cardiac procedures. However, these covariates are generally not modifiable intraoperatively. Alternatively, blood pressure and HR are the most natural targets for such anesthetic intervention. The CPH analysis confirms existing paradigms for blood pressure management, particularly the danger of low blood pressure. However, the largest increases in the predicted Hazard Ratios across both surgery types were due to increases in Mean HR; although, we recognize the confidence intervals are also large.

In summary, mean HR, mean MAP, and I(HR,MAP) represent targets for cardiovascular interventions during surgery, although this will need to be examined in future studies. In a broader sense, this result indicates a parsimonious measure of coupling through MI should replace correlation given its clear potential to capture complex dependence, improve existing epidemiological modeling, and provide physiologically-motivated interventions for anesthesiologists. The clinical implications of this research warrant additional investigation before guidelines for intraoperative patient care strategies can be developed. However, the results of the LR analysis indicate that the inclusion of hemodynamic data into existing scoring algorithms for risk assessment has clear potential to improve the identification of at-risk patients and improve postoperative care. Collectively, the CPH and LR studies demonstrate clear potential for MI as an additional tool for clinical and epidemiological data analysis.

## Supporting information

**S1 File. Technical details related to the definition, interpretation, and estimation of mutual information from data are provided therein.**
(PDF)

## Author Contributions

**Conceptualization:** Hau-Tieng Wu, David B. Dunson.

**Data curation:** Alexander L. Young, Willem van den Boom, Rebecca A. Schroeder.

**Formal analysis:** Alexander L. Young, David B. Dunson.

**Funding acquisition:** David B. Dunson.

**Investigation:** Alexander L. Young.

**Methodology:** Alexander L. Young.

**Supervision:** Vijay Krishnamoorthy, Karthik Raghunathan, David B. Dunson.

**Validation:** Vijay Krishnamoorthy, Karthik Raghunathan.

**Visualization:** Alexander L. Young.

**Writing – original draft:** Alexander L. Young.

**Writing – review & editing:** Alexander L. Young, Vijay Krishnamoorthy, Karthik Raghu-nathan, Hau-Tieng Wu, David B. Dunson.

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
