## [Editor Report · Decision Letter 0]

20 Jan 2022

PONE-D-21-37101Mutual Information: Measuring Nonlinear Dependence in Longitudinal Epidemiological DataPLOS ONE

Dear Dr. Young,

Thank you for submitting your manuscript to PLOS ONE. After careful consideration, we feel that it has merit but does not fully meet PLOS ONE’s publication criteria as it currently stands. Therefore, we invite you to submit a revised version of the manuscript that addresses the points raised during the review process. The problem of using mutual information to discover nonlinearities is well addressed in the literature. The so called 'maximal information coefficient,' or MIC, is introduced in

Kinney, J.B. and Gurinder S. A. (2014) Equitability, mutual information, and the maximal information coefficient, PNAS 111:119

More discussion can be found in

David N. Reshef, Yakir A. Reshef, Hilary K. Finucane, Sharon R. Grossman, Gilean McVean, Peter J. Turnbaugh, Eric S. Lander, Michael Mitzenmacher, Pardis C. Sabeti (2011). Detecting Novel Associations in Large Data Sets, Science 334.

Moreover, there is an R package that allows computation of MIC, minerva.

The authors miss these critical references.

The paper is poorly written. For example, the Appendix is full of?

We look forward to receiving your revised manuscript.

Kind regards,

Eugene Demidenko, Ph.D.

Academic Editor

PLOS ONE

Journal Requirements:

2. Please include a complete ethics statement in the Methods section, including the name of the IRB and the approval number, and whether they approved the study or waived the need for approval.

“ALY was supported by the National Science Foundation (Award #1045153, Award 331 #1546130)

WvdB was supported by the National Institute of Environmental Health 332 Sciences (NIEHS) of the National Institutes of Health (NIH) (R01-ES017240)”

“ALY was supported by the National Science Foundation (Award #1045153, Award #1546130). WvdB was supported by the National Institute of Environmental Health Sciences (NIEHS) of the National Institutes of Health (NIH) (R01-ES017240). The funders had no role in study design, data collection and analysis, decision to publish, or preparation of the manuscript.”

 “ALY was supported by the National Science Foundation (Award #1045153, Award 331 #1546130)

WvdB was supported by the National Institute of Environmental Health 332 Sciences (NIEHS) of the National Institutes of Health (NIH) (R01-ES017240)”

---

## [Author Response · Author response to Decision Letter 0]

29 Jun 2022

Below, please find are responses to the issues identified in our initial submission of Mutual information: measuring nonlinear dependence in longitudinal epidemiological data. Prior to those details, please let me apologize for the delay in response. De-identifying and preparing our patient data took considerably more time than we expected, but we have been able to make it available with publication.

Reviewer Comments

1. The problem of using mutual information to discover nonlinearities is well addressed in the literature. The so called 'maximal information coefficient,' or MIC, is introduced in

Kinney, J.B. and Gurinder S. A. (2014) Equitability, mutual information, and the maximal information coefficient, PNAS 111:119

More discussion can be found in

David N. Reshef, Yakir A. Reshef, Hilary K. Finucane, Sharon R. Grossman, Gilean McVean, Peter J. Turnbaugh, Eric S. Lander, Michael Mitzenmacher, Pardis C. Sabeti (2011). Detecting Novel Associations in Large Data Sets, Science 334.

Moreover, there is an R package that allows computation of MIC, minerva.

The authors miss these critical references.

We thank the reviewer for supplying these valuable references which have been included in the revised version of the manuscript.

2. The paper is poorly written. For example, the Appendix is full of?

The writing has been revised in several areas throughout the manuscript with a small reordering of the material to streamline the exposition. The multiple occurrences of ?? in the supplementary material resulted from an issue with BibTex and has been addressed in the updated version.

Journal Requirements

The author affliation symbols have been updated in concordance with the journal guidelines. Figure and Table citations have been arranged accordingly as well. The PLoS One style package was used to generate the manuscript.

2. Please include a complete ethics statement in the Methods section, including the name of the IRB and the approval number, and whether they approved the study or waived the need for approval.

The following ethics statement has been added to the Methods Section:

The Duke Medicine Institutional Review Board for Clinical Investigations approved this study (approval no.: Pro0067683). After full review, the requirement for informed consent of this retrospective study was waived as the data were analyzed anonymously. 

“ALY was supported by the National Science Foundation (Award #1045153, Award 331 #1546130)

WvdB was supported by the National Institute of Environmental Health 332 Sciences (NIEHS) of the National Institutes of Health (NIH) (R01-ES017240)”

Here is our updated financial disclosure which has been added to the updated cover letter as well.

ALY was supported by the National Science Foundation (Award #1045153, Award 331 #1546130). WvdB was supported by the National Institute of Environmental Health 332 Sciences (NIEHS) of the National Institutes of Health (NIH) (R01-ES017240). The funders had no role in study design, data collection and analysis, decision to publish, or preparation of the manuscript.

“ALY was supported by the National Science Foundation (Award #1045153, Award #1546130). WvdB was supported by the National Institute of Environmental Health Sciences (NIEHS) of the National Institutes of Health (NIH) (R01-ES017240). The funders had no role in study design, data collection and analysis, decision to publish, or preparation of the manuscript.”

 “ALY was supported by the National Science Foundation (Award #1045153, Award 331 #1546130)

WvdB was supported by the National Institute of Environmental Health 332 Sciences (NIEHS) of the National Institutes of Health (NIH) (R01-ES017240)”

We have removed the acknowledgments section from the manuscript. Please see item 4 above regarding our amended funding statement.

 Here is our updated Data Availability Statement which has been added to the manuscript:

 The data used for this study has been anonymized and made available through the Duke University Library Digital Repository for Research Data, doi:10.7924/r45q52g2t. 

 A caption for the Supporting Information files has been added to the end of the manuscript.

Thank you again for considering our manuscript. We will be happy to address any additional changes or updates you require.

---

## [Editor Report · Decision Letter 1]

15 Jul 2022

PONE-D-21-37101R1Mutual information: measuring nonlinear dependence in longitudinal epidemiological dataPLOS ONE

Dear Dr. Alexander Young,

Thank you for submitting your manuscript to PLOS ONE. After careful consideration, we feel that it has merit but does not fully meet PLOS ONE’s publication criteria as it currently stands. Therefore, we invite you to submit a revised version of the manuscript that addresses the points raised during the review process.

Although your work looks impressive two major concerns have to be addressed before I initiate the review process.1. You did not address my concern on the existing method of estimation of mutual information via MIC. It is not sufficient to cite the MIC research papers. More discussion and comparison must be provided. What if your estimator is equivalent to or even worse than MIC?2. The way to compare via simulations is to pick a distribution, say a multivariate normal distribution, and show how the MSE of both estimators converge to the theoretical (closed form) counterpart as a function of the number of observations. I believe that these additions will significantly increase the value of your work, albeit require more work.  

We look forward to receiving your revised manuscript.

Kind regards,

Eugene Demidenko, Ph.D.

Academic Editor

PLOS ONE
---

## [Author Response · Author response to Decision Letter 1]

30 Aug 2022

1. You did not address my concern on the existing method of estimation of mutual information via MIC. It is not sufficient to cite the MIC research papers. More discussion and comparison must be provided. What if your estimator is equivalent to or even worse than MIC?

This is an excellent point, and we appreciated the opportunity to address it. We have expanded the discussion on estimators of statistical dependence including references to distance correlation. During this discussion, we note the emphasis on power in this related literature is of less import than minimizing error when estimating quantities from finite samples. We address this issue through an empirical study on mean squared error included in the supporting materials.

2. The way to compare via simulations is to pick a distribution, say a multivariate normal distribution, and show how the MSE of both estimators converge to the theoretical (closed form) counterpart as a function of the number of observations.

We agree with this point. However, aside from mutual information and distance correlation in a few analytically tractable cases, estimators of statistical dependence do not have closed form expressions. As such, we have included two separate empirical studies. 

In the first, we focus on independent data in three settings (Gaussian, uniform, and Cauchy distributions) where mutual information, MIC, and distance correlation are all zero. These results suggest that MI as estimated using the nearest neighbor approach has mean squared error approximately two orders of magnitude lower than MIC or distance correlation across all sample sizes considered. 

In the second, we consider the bivariate Gaussian case with non-zero correlation. In this setting, both mutual information and distance correlation have closed form expressions. MIC does not. Thus, instead of focusing on MSE, we provide empirical estimates of the mean and variance of each estimator for comparison. All methods exhibit comparable variance across the chosen sample sizes. When studying the empirical means, the fluctuations around the true values of MI and dCor are comparable indicating similar small sample bias whereas the sample mean of MIC does not appear to have converged by the maximum sample size considered.

These studies, which we believe provide compelling justification for our methodology, were included in the supporting materials. We are open to moving these results into the main paper pending input from the reviewers.

---

## [Decision Letter · Decision Letter 2]

2 Nov 2022

PONE-D-21-37101R2Mutual information: measuring nonlinear dependence in longitudinal epidemiological dataPLOS ONE

Dear Dr. Young,

Thank you for submitting your manuscript to PLOS ONE. After careful consideration, we feel that it has merit but does not fully meet PLOS ONE’s publication criteria as it currently stands. Therefore, we invite you to submit a revised version of the manuscript that addresses the points raised during the review process.

The two reviewers, especially the second reviewer, have important comments to be addressed. Please respond to every point raised.

We look forward to receiving your revised manuscript.

Kind regards,

Eugene Demidenko, Ph.D.

Academic Editor

PLOS ONE

Reviewers' comments:

Reviewer's Responses to Questions

**Comments to the Author**

1. If the authors have adequately addressed your comments raised in a previous round of review and you feel that this manuscript is now acceptable for publication, you may indicate that here to bypass the “Comments to the Author” section, enter your conflict of interest statement in the “Confidential to Editor” section, and submit your "Accept" recommendation.

Reviewer #1: (No Response)

Reviewer #2: (No Response)

2. Is the manuscript technically sound, and do the data support the conclusions?

Reviewer #1: Yes

Reviewer #2: Partly

3. Has the statistical analysis been performed appropriately and rigorously? 

Reviewer #1: No

Reviewer #2: No

4. Have the authors made all data underlying the findings in their manuscript fully available?

Reviewer #1: Yes

Reviewer #2: No

5. Is the manuscript presented in an intelligible fashion and written in standard English?

Reviewer #1: Yes

Reviewer #2: Yes

6. Review Comments to the Author

Reviewer #1: The authors present a study that motivates the use of mutual information (MI), as a statistical summary of data interdependence and claims to be a suitable alternative or addition to correlation for identifying relationships in data. As a case study, the use of MI in the analyses of epidemiologic data is considered, while providing a general introduction to estimation and interpretation. The utility is illustrated through a retrospective study relating intraoperative heart rate (HR) and mean arterial pressure (MAP). The findings show that postoperative mortality is associated with decreased MI between HR and MAP and improve existing postoperative mortality risk assessment by including MI and additional hemodynamic statistics.

On a positive note, this revised study received in its current form has a few merits: Good articulation with a fair introduction section and explanatory diagrams.

However, the document in its current form is quite limited and seems to baffle the readers of the critical care research community. To be precise the article needs to relook and dive more into the non-linear associations of MI among the vital signs of the data for mortality risk assessment in comparison to other statistical measures.

The perplexities (and few suggestions) of the reviewer regarding the whole range of quality dimensions to judge a research paper on mortality risk assessment worthy of being published in PLOS ONE are as follows.

1. The authors should justify their need to resort to the logistic regression technique as a baseline model for data analysis and why not other models, especially gradient-boosting decision trees and/or neural networks. A comparative analysis with at least decision trees is much appreciated.

2. Comparative experimental results among Pearson correlation and other robust correlation measures such as distance correlation, mutual information, and maximal information coefficient should be depicted to support the hypothesis.

3. Can the ratios and higher-order statistic relations (square, cubic, etc) in terms of the vital signs among the data be useful to be considered? What are the advantages and disadvantages?

4. The discussion section can highlight the related studies used to explore the statistical dependencies among the vital signs for mortality risk assessment. The reviewer recommends the authors consult and cite the below-mentioned references. The most important must also be discussed, not just mentioned in a list of references.

5. Authors must enhance their experiments and try to present results based on evidence. Reporting AUROC, and MSE as performance metrics for mortality risk prediction is not fully correct. Reporting AUPRC, and PPV might add clinical significance. Even, further analysis of the variation of β on Fβ measure can serve the purpose to some extent in terms of robustness and significance.

References to be cited:

1. W. Lin, J. Ji, Y. Zhu, M. Li, J. Zhao, F. Xue, Z. Yuan, PMINR: pointwise mutual information-based network regression–with application to studies of lung cancer and Alzheimer's disease, Front. Genet. 11 (2020) 1043.

2. L. Lu, X. Ren, C. Cui, Y. Luo, M. Huang, Tensor Mutual Information and its Applications, Concurrency and Computation: Practice and Experience, 2020 e5686.

3. Cook NR. Use and misuse of the receiver operating characteristic curve in risk prediction. Circulation 2007;115:928–35.

4. Cook NR. Statistical evaluation of prognostic versus diagnostic models: beyond the ROC curve. Clin Chem 2008;54:17–23.

5. N. Nesaragi, S. Patidar, V. Aggarwal, Tensor learning of pointwise mutual information from EHR data for early prediction of sepsis, Computers in Biology and Medicine, vol. 134, pp.104430, 2021.

6. N. Nesaragi, S. Patidar, T. Veerakumar, A Correlation Matrix-based Tensor Decomposition Method for Early Prediction of Sepsis from Clinical Data, Biocybernetics and Biomedical Engineering, vol. 41(3):10131024, 2021.

7. N. Nesaragi and S. Patidar, Early prediction of sepsis from clinical data using ratio and power-based features," Critical Care Medicine, vol. 48, no. 12, pp. e1343-e1349, 2020.

8. R. Krishnan, G. Sivakumar, P. Bhattacharya, Extracting decision trees from trained neural networks, 1999. Pattern recognition 32.

Reviewer #2: The authors explain the concept of mutual information (MI) and propose to use it as a measure of association between variables. They apply it to a data set on postoperative mortality.

On MI:

I agree that R^2 is not a good measure if relationships are highly nonlinear. As long as the relation is monotone, a rank-correlation can be used instead. It is certainly useful to look for better and more general measures. The authors propose to use MI, which quantifies association more rubustly. They give an example with the gapminder data. For the relation between SBP and cholesterol, MI is clearly better than R^2. However, I find it difficult to believe that MI is larger than for BMI and cholesterol, which show a much stronger association in the scatterplot. This suggests that R^2 is a better measure in case the association is fairly linear. I would like to see some discussion on this by the authors.

In the supplementary material, three statistics are compared using MSE. I am not convinced that MI is better. MI, MIC and cCor are different quantities. Even if they are standardized, a value of say 0.1 can have a different meaning for each of them. There is no standardized way to compare them.

This is also seen in Figure 2, where data with correlation is compared. Estimates agree fairly well with true values, except maybe for N=100. KSG has smaller values, but that doesn't mean it gives a better reflection of correlation. I don't see a justification for the claim "one expects KSG to have lower MSE when compared to MIC and dCor in this setting as well". The lower variance for KSG is to be expected given the lower average value.

The data set:

1) Do the authors expect that the relation between hemodynamic variables and outcome remains the same if HR and MAP are modified via an intervention?

2) The authors consider a Cox (CPH) model and a logistic regression (LR). For the LR, the model including hemodynamic variables improves with respect to AUC (Table 4). Figure 5 shows two individuals; for the second one, MI is clearly better than R^2. However, the authors do not report to what extent MI improve upon HR and MAP itself with respect to AUC. Also, I would liek to see the AUC for the CPH model included.

3) Several selections were made. Why were those with ASA score 5 excluded? In Table 2 I see that the number with ASA 1 or 2 is even smaller, while these individuals were not excluded.

Excluding those with fewer than 240 HR/MAP values gives a risk of selection bias. It may be that these individuals have shorter follow-up due to death. The authors include the number of measurements as additional variable (in CPH only), but I don't see these results reported.

4) Were there any right censored individuals within 30 days?

5) Correlation does show strong relation with mortality in Fig 6D, but the authors write " does not display a strong association with mortality"

6) What is the biological reason to include MI in the model? Why would MI relate to mortality?

7) If two variables have a clear trend in individuals over time, it may be better first to remove that trend. It will give a strong correlation, which may have little biological relevance. It may be better to quantify the correlation after removal of the trend.

Supplementary material:

0 log 0 is zero mathematically, it is not a convention

I don't understand why the entropies for continuous distributions do not carry the same amount of information and why " an infinite number of entries after the decimal" is a problem. They are integrals that can be approximated as accurately as on wishes (e.g. via Riemann sums).

How is k chosen?

Several typos in text and formulas (also in the main text). Two examples: "d" in supplement form (2) and log2(3)-2/3

7. PLOS authors have the option to publish the peer review history of their article (what does this mean?). If published, this will include your full peer review and any attached files.

Reviewer #1: No

Reviewer #2: **Yes: **R.B. Geskus

---

## [Author Response · Author response to Decision Letter 2]

3 Apr 2023

Dear Dr. Demidenko,

We thank you for considering our manuscript and for the additional detailed comments of your reviewers. Their comments would, indeed, strengthen the results for the perioperative risk assessment community. However, our focus in this article is to demystify the use of mutual information rather than providing optimal risk assessment tools. We have made comments throughout the manuscript to clarify this point and to address some of the additional remarks regarding clarity and typographical issues. Detailed comments to each point are provided below.

Reviewer 1 Comments

1. The authors should justify their need to resort to the logistic regression technique as a baseline model for data analysis and why not other models, especially gradient-boosting decision trees and/or neural networks. A comparative analysis with at least decision trees is much appreciated.

We greatly appreciate this comment. A broad study of the different modeling frameworks for mortality prediction is clearly of interest to the clinical care community. However, our aim in this article is to introduce and demystify Mutual Information. As such, we have elected to focus on building intuition and interpretability of MI. The examples were chosen to demonstrate improvements to existing models which are attainable with the inclusion of nonlinear dependence as measured by MI.

2. Comparative experimental results among Pearson correlation and other robust correlation measures such as distance correlation, mutual information, and maximal information coefficient should be depicted to support the hypothesis.

The absolute value of the Pearson correlation is included already. The use of signed Pearson correlation does not alter the results, and we have added a comment on this issue to the manuscript. For the remaining methods, we would like to reemphasize the analysis in the supplemental materials on the estimation of MI vs other methods. Additional details on that analysis are discussed below. 

3. Can the ratios and higher-order statistic relations (square, cubic, etc) in terms of the vital signs among the data be useful to be considered? What are the advantages and disadvantages?

Standard deviation of Heart Rate and Mean Arterial Pressure were included. By using generalized additive models, we have thus included quadratic statistics, though we admit that these results are not clearly interpretable from the presented graphs. Ratios and higher-order terms were not considered. We agree that one could consider such terms as a method of improving model fit at the cost of model complexity and the risk of overfitting. However, as we discussed in the introduction on R2, the inclusion of such terms is often ad hoc. We have expanded on this observation in the introduction.

4. The discussion section can highlight the related studies used to explore the statistical dependencies among the vital signs for mortality risk assessment. The reviewer recommends the authors consult and cite the below-mentioned references. The most important must also be discussed, not just mentioned in a list of references.

We greatly appreciate the recommend references. In particular, the sequence of studies by Neseragi et al are of central interest to this article as they provide an additional example motivating the use of MI over correlation and ratio-based statistics. We have added detail on these results to our introduction.

5. Authors must enhance their experiments and try to present results based on evidence. Reporting AUROC, and MSE as performance metrics for mortality risk prediction is not fully correct. Reporting AUPRC, and PPV might add clinical significance. Even, further analysis of the variation of β on Fβ measure can serve the purpose to some extent in terms of robustness and significance.

Again, we would like to emphasize that the motivation of the article is to introduce MI and its utility to the perioperative risk assessment community, not to identify optimal models. As such, we believe keeping the model assessment parsimonious to emphasize the interpretation of MI earlier in the article is optimal.

Reviewer 2 Comments

 On MI:

1. I agree that R^2 is not a good measure if relationships are highly nonlinear. As long as the relation is monotone, a rank-correlation can be used instead. It is certainly useful to look for better and more general measures. The authors propose to use MI, which quantifies association more robustly. They give an example with the gapminder data. For the relation between SBP and cholesterol, MI is clearly better than R^2. However, I find it difficult to believe that MI is larger than for BMI and cholesterol, which show a much stronger association in the scatterplot. This suggests that R^2 is a better measure in case the association is fairly linear. I would like to see some discussion on this by the authors.

This raises a very interesting point on the dependency of tuning k in the nearest neighbor method. Choosing k=10 results in a relationship which is consistent with the reviewers assessment. We have discussed this sensitivity to parameter tuning as a weakness of MI estimation and provided complete result for the gapminder data using k=10 in the supplementary material.

2. In the supplementary material, three statistics are compared using MSE. I am not convinced that MI is better. MI, MIC and cCor are different quantities. Even if they are standardized, a value of say 0.1 can have a different meaning for each of them. There is no standardized way to compare them.

This is also seen in Figure 2, where data with correlation is compared. Estimates agree fairly well with true values, except maybe for N=100. KSG has smaller values, but that doesn't mean it gives a better reflection of correlation. I don't see a justification for the claim "one expects KSG to have lower MSE when compared to MIC and dCor in this setting as well". The lower variance for KSG is to be expected given the lower average value.

This is an excellent point. We have expanded the analysis in the supplement to also compute relative error, as measured by the ratio of bias and root mean squared error. In all cases considered, KSG has the lowest relative error. 

On the data set:

1. Do the authors expect that the relation between hemodynamic variables and outcome remains the same if HR and MAP are modified via an intervention?

MAP (and HR in some hospitals) are already modified by an intervention. How one could intervene on I(HR,MAP) is unknown at present. 

2. The authors consider a Cox (CPH) model and a logistic regression (LR). For the LR, the model including hemodynamic variables improves with respect to AUC (Table 4). Figure 5 shows two individuals; for the second one, MI is clearly better than R^2. However, the authors do not report to what extent MI improve upon HR and MAP itself with respect to AUC. Also, I would like to see the AUC for the CPH model included.

Again, we would like to maintain a focus on MI and the interesting connection between decreased dependence and mortality risk rather than model validation.

3. Several selections were made. Why were those with ASA score 5 excluded? In Table 2 I see that the number with ASA 1 or 2 is even smaller, while these individuals were not excluded.

ASA 5 patients are critically ill and require surgical intervention. We removed those cases from the analysis to focus on settings where surgical status was not dictated by an immediate medical need. We have added a statement on this issue to the article.

Excluding those with fewer than 240 HR/MAP values gives a risk of selection bias. It may be that these individuals have shorter follow-up due to death. The authors include the number of measurements as additional variable (in CPH only), but I don't see these results reported.

Hazard ratios for the number of measurements in each surgery type have been added with a brief comparison of these results with the existing discussion on hazard ratios.

4. Were there any right censored individuals within 30 days?

All cases where the patient survived for more than one year were censored.

5. Correlation does show strong relation with mortality in Fig 6D, but the authors write " does not display a strong association with mortality"

This is a fair point. We have updated the language to note association of correlation with outcome in the cardiac cases.

6. What is the biological reason to include MI in the model? Why would MI relate to mortality?

The dependence of HR and MAP has been considered in other studies. In particular, the rate pressure product (RPP) has been considered in other articles including the Krishnamoorthy et al. article in our references. When discussing other means of measuring this association between mixed second moments, the mathematical side of our research team suggested MI given issues on nonlinear behavior of RPP. At present, tt is unclear how MI may relate to mortality through a clear biological mechanism, and we can only offer circumspection which is not justifiable to include in the article. Nonetheless, we do believe this finding could be of interest and have added a comment to the manuscript.

7. If two variables have a clear trend in individuals over time, it may be better first to remove that trend. It will give a strong correlation, which may have little biological relevance. It may be better to quantify the correlation after removal of the trend.

This is a natural point, but removing a trend could leave a nonlinear association in the residuals which would be missed by correlation. 

On Supplementary material:

1. 0 log 0 is zero mathematically, it is not a convention.

Informed minds can disagree on the technicalities of this point. The function x log(x) is undefined at x=0, hence our point on the convention of taking 0log(0) = 0, which is a common remark in the information theoretic literature. However, the limit as of x log(x) is zero as x approaches zero and it is thus natural to take 0 log(0) = 0 and treat it as a definition. These details do not add substance to the discussion, and we have elected to remove the language regarding convention in line with the reviewer's remarks.

2. I don't understand why the entropies for continuous distributions do not carry the same amount of information and why " an infinite number of entries after the decimal" is a problem. They are integrals that can be approximated as accurately as on wishes (e.g. via Riemann sums).

This is an excellent question, and the Cover text, which we have cited in the supplement, provides a complete answer. 

Briefly, we will provide some insight here. As a counterpoint, first consider the discrete case first for a finite set {1,…,N} of possible outcomes with associated probabilities p1, …, pN. The entropy, as measured in bits, provides information on the average number of yes/no answers one needs answered on average to identify a random sample from {1,…,N} based on the probabilities. If pj=1 for some j, the entropy is zero since the outcome would be known a priori. If the distribution is uniform, log2(N) yes/no questions are needed to identify the outcome, which is obtainable by splitting the possible outcomes into approximately even sized disjoint sets and querying if the realization is in one of these sets. One can the proceed sequentially to finer subsets until the realization is identified. 

If the outcome space is continuous, the potential realizations of the random outcome are irrational numbers with probability 1, hence our comment about the infinite number of entries after the decimal. One could elect to discretize the continuous distribution into partitions at which point identifying which partition contains the sample is almost equivalent to the discrete case above. The corresponding Riemann sum approximation to the integral provides the associated entropy of the discretized case minus an additional term of the form log (partition size) which accounts for the discretization error. As one sends the partition size to 0, the log (partition size) term is formally dropped leaving the differential entropy. If one were to keep this term the entropy discretized approximation would diverge to infinity as the partition size shrunk to zero.

We have attempted to improve this explanation by noting that the outcome must be irrational, hence impossible to store in finite memory. Additional details on the delicate nature of this point are directed to the reference by Cover et al.

3. How is k chosen?

k was chosen for to balance faster estimation with minimal sampling variability as data were jittered. We have added this point to the manuscript.

---

## [Editor Report · Decision Letter 3]

12 Apr 2023

Mutual information: measuring nonlinear dependence in longitudinal epidemiological data

PONE-D-21-37101R3

Dear Dr. Young,

We’re pleased to inform you that your manuscript has been judged scientifically suitable for publication and will be formally accepted for publication once it meets all outstanding technical requirements.

Kind regards,

Eugene Demidenko, Ph.D.

Academic Editor

PLOS ONE

Additional Editor Comments (optional):

The authors fully addressed the comments -- the paper can be published now. 
---

## [Editor Report · Acceptance letter]

17 Apr 2023

PONE-D-21-37101R3 

Mutual information: measuring nonlinear dependence in longitudinal epidemiological data 

Dear Dr. Young:

I'm pleased to inform you that your manuscript has been deemed suitable for publication in PLOS ONE. Congratulations! Your manuscript is now with our production department. 

Kind regards, 

on behalf of

Dr. Eugene Demidenko 

Academic Editor

PLOS ONE